# Meron-Mediated Phase Transitions in Quasi-Two-Dimensional Chiral Magnets with Easy-Plane Anisotropy: Successive Transformation of the Hexagonal Skyrmion Lattice into the Square Lattice and into the Tilted FM State

**DOI:** 10.3390/nano14181524

**Published:** 2024-09-20

**Authors:** Andrey O. Leonov

**Affiliations:** 1International Institute for Sustainability with Knotted Chiral Meta Matter, Kagamiyama, Higashihiroshima 739-8511, Hiroshima, Japan; leonov@hiroshima-u.ac.jp; 2Department of Chemistry, Faculty of Science, Hiroshima University Kagamiyama, Higashihiroshima 739-8526, Hiroshima, Japan

**Keywords:** skyrmion, bimeron, chiral magnets, induced Dzyaloshinskii–Moriya interaction

## Abstract

I revisit the well-known structural transition between hexagonal and square skyrmion lattices and subsequent first-order phase transition into the tilted ferromagnetic state as induced by the increasing easy-plane anisotropy in quasi-two-dimensional chiral magnets. I show that the hexagonal skyrmion order first transforms into a rhombic skyrmion lattice, which, adjusts into a perfect square arrangement of skyrmions (“a square meron-antimeron crystal”) within a narrow range of anisotropy values. These transitions are mediated by merons and anti-merons emerging in the boundaries between skyrmion cells; energetically unfavorable anti-merons annihilate, whereas pairs of neighboring merons merge. The tilted ferromagnetic state sets in via mutual annihilation of oppositely charged merons; as an outcome, it contains bimeron clusters (chains) with the attracting inter-soliton potential. Additionally, I demonstrate that domain-wall merons are actively involved in the dynamic response of the square skyrmion lattices. As an example, I theoretically study spin–wave modes and their excitations by AC magnetic fields. Two found resonance peaks are the result of the complex dynamics of the domain-wall merons; whereas in the high-frequency mode the merons rotate counterclockwise, as one might expect, in the low-frequency mode merons are instead created and annihilated consistently with the rotational motion of the domain boundaries.

## 1. Introduction

In magnetic compounds with broken inversion symmetry, the chiral crystal lattice induces a specific asymmetric exchange coupling, the so-called Dzyaloshinskii–Moriya interaction (DMI) [1,2]. Within a continuum approximation for magnetic properties, the DMI is expressed by Lifshitz invariants (LI), that is, the energy terms involving the first derivatives of the magnetization m with respect to the spatial coordinates $x_{k}$:(1)$$L_{i,j}^{\left(k\right)}=m_{i}\partial m_{j}/\partial x_{k}-m_{j}\partial m_{i}/\partial x_{k}.$$
Depending on the crystal symmetry, certain combinations of the Lifshitz invariants can contribute to the magnetic energy of the material [3,4].

In a general case of cubic helimagnets, such as the itinerant magnets MnSi [5,6] and FeGe [7] and the Mott insulator Cu_2_OSeO_3_ [8,9], the DMI reduces to the following concise form [1,3]:(2)$$\begin{array}{c} w_{D}=L_{z,y}^{\left(x\right)}+L_{x,z}^{\left(y\right)}+L_{y,x}^{\left(z\right)}=m\cdot \mathrm{rot}m. \end{array}$$

In polar magnets with $C_{nv}$ symmetry, such as GaV_4_S_8_ and GaV_4_Se_8_ [10,11], the DMI energy density
(3)$$w_{D}=m_{x}\partial_{x}m_{z}-m_{z}\partial_{x}m_{x}+m_{y}\partial_{y}m_{z}-m_{z}\partial_{y}m_{y}$$
does not include LIs along the high-symmetry *z* axis.

DMI of the same functional form (Equation 3) is also induced in multilayered structures due to breaking of the inversion symmetry at interfaces, as occurs in, e.g., PdFe/Ir (111) [12]. Such artificial systems, enabled by the possibility of stacking, are extremely versatile as regards the choice of the magnetic, non-magnetic, and capping layers.

LIs (Equation 1) are indispensable to overcoming the constraints of the Hobart–Derrick theorem [13,14] and to yielding a set of competing modulated phases (generally, multidimensional) as well as countable solitons in the phase diagrams specific to different crystallographic classes.

In cubic helimagnetsm the rotational terms (Equation 2) are present along all three spatial directions; therefore, the modulated phases and solitons are expected to be truly three-dimensional (3D). For example, magnetic hopfions are torus-shaped solitons embedded into a homogeneously magnetized background and characterized by linked pre-images [15,16]. Generally, the magnetic phases in cubic chiral magnets develop additional twists involving all LIs near the surfaces (so-called “surface twists”), which are known to be essential for their thermodynamic stability [17]. Recently, skyrmion lattice states (SkL) and isolated skyrmions (ISs) have been discovered in bulk crystals of chiral magnets near the magnetic ordering temperatures [6,7] and in nanostructures with confined geometries over larger temperature regions [18,19].

Skyrmions generate enormous interest due to the prospect of their applications in information storage and processing [20,21,22]. Indeed, skyrmions are topologically protected [23], have the nanometer size [24], and can be manipulated by electric currents [25,26]. Skyrmions are also interesting objects for magnonics, e.g., collective spin dynamics within SkLs exhibit two spin–wave modes with the clockwise (CW) and counterclockwise (CCW) rotation of skyrmions for the in-plane AC magnetic field as well as one breathing mode for the out-of-plane AC magnetic field [27].

For $C_{nv}$ symmetry, only modulated magnetic structures with wave vectors perpendicular to the polar axis are favored by the DMI (Equation 3), and as such represent 2D motifs of the magnetization. Nevertheless, the phase diagrams constructed for such quasi-two-dimensional chiral magnets are far from being simple. Figure 1a shows the well-known phase diagram for chiral magnets with easy-plane anisotropy (EPA) [28,29]. In addition to homogeneous field-polarized and tilted states (Figure 1b,c), the phase diagram features one-dimensional cycloids and elliptical cones (Figure 1d,e). Moreover, it implies that two types of skyrmion orderings—square and the hexagonal SkLs—are stable even though the system does not have any anisotropy axis within the plane (Figure 1f,g). Notice that in the phase diagram of chiral magnets with DMI (Equation 2) and easy-plane uniaxial anisotropy, the conical phase with the wave vector along the field is the only stable modulated state [30], which makes the phases in Figure 1 energetically less favorable.

Certain phase transitions in the phase diagram in Figure 1a are well-understood: (i) the first-order phase transition between the cycloid and the hexagonal SkL (line *d*–*c* in Figure 1a) occurs via ruptures of the the cycloidal state called meron pairs, which acquire the energetic advantage above this critical field (for details, see, e.g., [31,32,33]); (ii) the second-order phase transition between a skyrmion lattice and the field-polarized FM state (line *a*–*b* in Figure 1a) occurs via the infinite expansion of the lattice period [34,35]; (iii) the elliptical cone and the tilted FM state gradually align along the field at line *a*–*e*.

Other phase transitions in the phase diagram are less understood. We can only anticipate that the anisotropy-driven phase transitions between hexagonal and square SkLs as well as between the square SkLs and 1D elliptical cones are of the first order [28,29], as they occur between topologically incompatible phases.

In the present manuscript, I re-examine the mentioned unclear transitions. I show that the reorientation transition between the hexagonal and square skyrmion arrangements with increasing anisotropy value occurs via distorted (rhombic) SkLs (Section 3). The distorted SkLs represent the global minima of the system, and gradually transit into the square SkL. The hexagonal SkL remains almost intact until its energy minimum disappears during this first-order phase transition. I underline the decisive role of merons and anti-merons formed within the boundary regions between skyrmion cells in SkLs. Anti-merons with negative topological charge density are shared by two adjacent skyrmion cells and bear positive energy density. Because there are three such anti-merons within one hexagonal unit cell, the square cell with just two anti-merons becomes energetically more favorable, even though the skyrmion packing density slightly decreases. During the structural transition, two corner merons with positive topological charge density merge and annihilate one unfavorable anti-meron. Two such annihilation events at the opposite cell boundaries signify the reorientation transition from hexagonal to square skyrmion order.

The phase transition between the square SkL and the tilted ferromagnetic state is also meron-mediated (Section 5). During this process, merons and anti-merons mutually collapse. Because the corner merons are shared by four neighboring skyrmion cells and the anti-merons by two skyrmion cells, there are two anti-merons and one meron per unit cell. After collapse, the remaining anti-meron couples with the central anti-meron and forms a localized state known as a bimeron [36]. While both anti-merons have negative topological charge, they have opposite polarity. As a result, the homogeneous state contains some finite number of bimerons (one per unit cell), just as the field-polarized FM state would host isolated axisymmetric ISs. Hence, the findings of the present paper shed new light on the phase transitions among different phases and imply that merons are important drivers guiding the whole process.

In addition, I study the collective spin dynamics of merons within the square skyrmion lattice (Section 4). I find two spin-wave resonances: (i) in the high-frequency mode, the central anti-meron performs counterclockwise rotation; (ii) in the low-frequency mode, the central anti-meron is virtually immobile, but the domain boundary approaches it with each side sequentially; this rotation of the domain-wall (DW) network is accompanied by creation and annihilation of DW merons.

## 2. Phenomenological Model

The magnetic energy density of a two-dimensional noncentrosymmetric ferromagnet can be written as the sum of the exchange, DMI (Equation 3), Zeeman, and anisotropy energy contributions, correspondingly:(4)$$w\left(m\right)=\sum_{i,j}{\left(\partial_{i}m_{j}\right)}^{2}+w_{D}-m\cdot h-k_{u}m_{z}^{2}.$$
Here, non-dimensional units are introduced to make the results more general and allow them to be directly mapped to any material system. Spatial coordinates x are measured in units of the characteristic length of modulated states $L_{D}$, A>0 is the exchange stiffness, *D* is the Dzyaloshinskii constant, and $k_{u}$ is the non-dimensional anisotropy constant which leads to the easy-plane magnetization, i.e., $k_{u}<0$.
(5)$$\begin{array}{cc} \; & L_{D}=A/D,k_{u}=K_{u}A/D^{2}\\ \; & h=H/H_{0},H_{0}=D^{2}/A\left|M\right| \end{array}$$
In addition, h is the magnetic field applied along the *z* axis. The magnetization vector m(x,y) is normalized to unity.

Alternatively, the length scale can be measured in units of λ:(6)$$\lambda =4\pi L_{D}$$
which is the period of the spiral state for zero magnetic field (e.g., 18 nm for bulk MnSi or 60 nm for Cu_2_OSeO_3_ [8,37]). In actual simulations, length is measured in units of $L_{D}$ (Equation 5). Dividing by 4π provides the length scale in units of λ, which enables a direct comparison with a specific material system. Both length scales are used throughout this paper.

Considering a 2D film of a ferromagnetic material on the xy-plane using periodic boundary conditions (pbc), the value of the field is kept constant h=0.5, whereas the value of the anisotropy constant $k_{u}$ is changed to address the aforementioned phase transitions between modulated phases.

The influence of dipole–dipole interactions is neglected due to the magnetic charges formed within different states with a Neel-like type of magnetization rotation. It is assumed that the DM interactions suppress demagnetization effects and are the main driving force leading to magnetization rotation and equilibrium periodicity. Moreover, in this case the shape anisotropy represents an additional correction of the easy-plane anisotropy. The influence of dipole–dipole interactions on the effects found in the present manuscript will be considered elsewhere.

The MuMax3 software package (version 3.10) is used as the primary numerical tool to minimize the functional (Equation 4), which calculates the magnetization dynamics by solving the Landau–Lifshitz–Gilbert (LLG) equation with the finite difference discretization technique [38]. To double-check the validity of obtained solutions, this paper also uses self-designed numerical routines which are explicitly described in, e.g., [39], and hence are omitted here.

All structures are minimized on a 256×256×1 grid. To check the stability of different skyrmion orderings, the energy density (Equation 4) is computed for different ratios of the grid spacings $\Delta_{y}$ and $\Delta_{x}$ (called cell sizes in mumax3, Figure 2a); $\Delta_{z}=0.1$ remains the same in all simulations. The axisymmetric distribution of the magnetization within skyrmion cores is preserved during this minimization procedure. Thus, varying lattice spacings lead to rearrangement of the constituent skyrmion cores spanning all possible lattice orders.

Figure 2a shows the centered rectangular unit cell used for computation of the skyrmion orderings. To characterize the degree of SkL deformations, the following lengths and angles are introduced, consistent with the square and the hexagonal SkLs (blue circles correspond to skyrmion centers, Figure 2b): (i) within the square SkL, $b_{1}=b_{2}=\sqrt{2}a$, $\gamma =45^{\circ }$; (ii) within the hexagonal SkL, $b_{1}=a=b_{2}/\sqrt{3}$, $\gamma =30^{\circ }$.

As an example, Figure 2c shows the color plot of the energy density depending on the cell sizes for $h=0.5,\;k_{u}=0.6785$. The red and black lines highlight the grid spacings for the square ($\Delta_{x}=\Delta_{y}$) and hexagonal ($\Delta_{x}=\Delta_{y}\sqrt{3}$) skyrmion lattices. The energy minimum corresponds to a slightly distorted SkL with lattice parameters $\Delta_{x}^{min}=0.161,$$\Delta_{y}^{min}=0.095$, which is very close to the hexagonal skyrmion ordering and corresponds to $b_{1}=N_{x}\Delta_{x}^{min}/4\pi =3.28,\;b_{2}=1.94$ and $\gamma =30.54^{\circ }$. The other energy minimum is reached for the interchanged lattice parameters $\Delta_{y}^{min}=0.161,\;\Delta_{x}^{min}=0.095$ when the energy contour plot is mirrored with respect to the red line. In the following, only the lower part of the total energy density distribution is considered (compare with the color plot in Figure 5g,h).

The energy density in Figure 2c is computed as follows:$$\varepsilon =\left(1/V\right)\int w\left(x,y\right)dV,\;V=N_{x}N_{y}N_{z}\Delta_{x}\Delta_{y}\Delta_{z}$$
where *V* is the volume of the unit cell in Figure 2a, $N_{x}=256,N_{y}=256$, and $N_{z}=1$. To highlight the topology of the energy surface in the direct vicinity of the energy minimum, the color plot discerns the energy range from the minimal energy value to $\varepsilon_{min}+3\times 10^{-5}$.

## 3. Reorientation Transition between Hexagonal and Square Skyrmion Arrangements

Figure 3 shows a series of energy “fingerprints” for the increasing value of the negative EPA $k_{u}$. A new minimum corresponding to a distorted SkL is clearly visible forming in (a) for $k_{u}=-0.6790$. It gradually deepens (Figure 3b,c) and equals the energy minimum of the hexagonal SkL at $k_{u}^{0}=-0.6798$. After this anisotropy value, the hexagonal SkL becomes a metastable state. At the value $k_{u}=-0.6817$, the local energy minimum of the hexagonal SkL disappears and the global minimum corresponds to a square lattice, which becomes fully shaped at $k_{u}=-0.6826$. Thus, in the anisotropy range $k_{u}\in \left[-0.6817,-0.6788\right]$ there are coexisting solutions for two SkLs, which underlie the first-order phase transition.

The lattice parameters for both skyrmion orders are shown in Figure 4a. The red (blue) solid lines correspond to the hexagonal (distorted) SkL, which is the global minimum of the system, whereas the dotted lines correspond to metastable solutions. The dashed vertical lines indicate the hysteresis behavior of the reorientation transition and highlight the limiting anisotropy values of the loop as well as the critical value of the anisotropy $k_{u}^{0}$. The dark blue line corresponds to the square SkL with equal parameters $b_{1}=b_{2}$. The angle γ (Figure 4b) changes almost linearly until it reaches the value $\gamma \approx 45^{\circ }$ when the anisotropy value $k_{u}^{0}$ is surpassed, and stays almost intact below this point ($\gamma \approx 30^{\circ }$).

The underlying reason for this phase transition can be elucidated from the energy density distributions w(x,y) as well as the topological charge densities $\rho_{Q}$ within different skyrmion arrangements. Figure 5 features hexagonal and distorted SkLs for $k_{u}^{0}=-0.6798$ when the corresponding energy minima are equal.

Figure 5a,b exhibit the color plots for the energy density ε to show all four solutions; solutions above the red line are rotated by $90^{\circ }$ with respect to the solutions below the red line. Figure 5b zooms the energy landscape in the direct vicinity of the energy minima $\varepsilon \in [\varepsilon_{min},\varepsilon_{min}+3\times 10^{-6}]$. Iy should be noted that these color plots are just top views of the 3D energy density surfaces (Figure 5c).

For the relatively large value of the easy-plane anisotropy at $k_{u}^{0}$, the role of merons formed within the domain boundaries between skyrmion cells becomes, paramount as will be seen later. According to [28], merons emerge due to the overlap of neighboring skyrmions.

Let us first scrutinize the internal structure of the hexagonal SkLs. The anti-merons (highlighted by dashed black circles in Figure 5d–f) have positive energy density (Figure 5d) and negative topological charge density (Figure 5e,f). They also have negative vorticity (Figure 5f). Figure 5e,f shows the topological charge density, with Figure 5e zoomed-in on the interval $\rho_{Q}\in \left[-0.05,0.05\right]$. Figure 5f shows just two unit cells, with the in-plane components of the magnetization vectors as black arrows. Because each anti-meron is shared by two adjacent skyrmion cells (highlighted by dashed white hexagons), there are three anti-merons per unit cell. The central vortex also represents an anti-meron. Although it has positive vorticity, its negative polarity endows it with negative topological charge.

Merons (highlighted by the dashed yellow circles in Figure 5d–f) exhibit positive topological charge density (Figure 5e,f) and negative energy density (Figure 5d), which is the outcome of their positive vorticity and polarity. Because merons are located in the corners, they are shared by three neighboring unit cells, resulting in two merons per unit cell.

In the distorted SkL (Figure 5g–i), two corner merons approach each other and eliminate one anti-meron located between them. The two remaining merons are easily discernible (highlighted by dashed yellow ellipses in Figure 5h,i). The energetic advantage of the distorted SkL due to the collapse of the anti-merons is counterbalanced by slightly higher skyrmion density. At larger anisotropy values, two merons merge into one and the lattice becomes a perfectly square arrangement of skyrmions, as shown in Figure 4c (last panel). The unit cell includes one meron and two anti-merons.

The results of the current section imply that at high anisotropy values the internal structure of the SkL can be considered from the point of view of interacting merons confined within the stretched domain boundaries (Figure 4c). Such a network of merons has been dubbed a square vortex–antivortex crystal [28]. Alternatively, it could be called a ‘square meron–antimeron crystal’.

## 4. Spin–Wave Modes of the Square SkL

It is instructive to investigate the dynamics of the meron–antimeron crystal under external oscillating magnetic fields and to deduce whether it becomes different as compared with the excitation effects in hexagonal SkLs [27].

Thus, the collective spin dynamics of meron crystals are studied for $k_{u}=-0.685,$
h=0.5 following the numerical procedure explicitly described in [27].

First of all, the discretized version of Equation (Equation 4) is adapted, with the same energy terms:(7)$$\begin{array}{cc} \; & w\left(S\right)=J\;\sum_{<i,j>}\left(S_{i}\cdot S_{j}\right)-\sum_{i}H\cdot S_{i}-K_{u}{\left(S_{i}\cdot \hat{z}\right)}^{2}\\ \; & -D\;\sum_{i}\left(S_{i}\times S_{i+\hat{x}}\cdot \hat{y}-S_{i}\times S_{i+\hat{y}}\cdot \hat{x}\right). \end{array}$$
We consider classical spins $S_{i}$ of unit length on a square two-dimensional lattice, where <i,j> denote pairs of nearest-neighbor spins. All calculations are performed for spin systems with 104×104 sites under pbc which include four unit cells. The spin configuration is minimized with respect to the period, and corresponds to the energy minimum for the chosen control parameters. The DMI constant D=Jtan(2π/λ) defines the period of modulated structures λ (Equation 6). In what follows, we use J=−1 and the DMI constant is set to 0.3249, i.e., λ≈20.

Next, the LLG equation is solved numerically under time-dependent AC magnetic fields using the fourth-order Runge–Kutta method. The equation is provided by
(8)$$\frac{\partial S_{i}}{\partial t}=-\frac{1}{1+\alpha_{G}^{2}}\left[S_{i}\times H_{i}^{eff}+\frac{\alpha_{G}}{S}S_{i}\times \left(S_{i}\times H_{i}^{eff}\right)\right],$$
where $\alpha_{G}$ is the dimensionless Gilbert damping coefficient. A rather small dimensionless damping parameter of $\alpha_{G}=0.01$ is used to ensure that all peaks in the imaginary part of the dynamical magnetic susceptibility are visible (Figure 6a). Here, $H_{i}^{eff}$ is a local effective field acting on the *i*-th spin $S_{i}$ and derived from the Hamiltonian $H_{i}^{eff}=-\partial w/\partial S_{i}$.

To study the microwave absorption spectra due to spin–wave resonances in the square SkL, in-plane δ-function pulses of magnetic field $h^{\omega }=0.1$ are applied at t=0, then the trace spin dynamics are traced. The absorption spectrum of the imaginary part of the dynamical susceptibility is Imχ(ω), calculated from the Fourier transformation of the magnetization $m=\left(1/N\right)\sum S_{i}\left(t\right)$. Figure 6a shows the imaginary part of the in-plane dynamical magnetic susceptibility in dependence on ω for the chosen control parameters. The calculated spectrum for $h^{\omega }$ parallel to the *y* axis exhibits two resonance peaks at $\omega_{1}=0.0025$ and $\omega_{2}=0.0107$.

To identify each spin–wave mode, the spin dynamics is traced by applying an oscillating magnetic field with a corresponding resonant frequency and amplitude $h^{\omega }=0.001$ (see Appendix A). Figure 6b,c shows the average components of the magnetization in each mode $\langle m_{x}\rangle ,\;\langle m_{y}\rangle ,\;\langle m_{z}\rangle$. Figure 6d displays the calculated time evolutions of the spins in the first mode. The in-plane projections of the spins are represented by black arrows, while the color plots are the topological charge distributions zoomed in the interval $\rho_{Q}\in \left[-0.05,0.05\right]$.

The high-frequency mode represents a CCW rotation of the central anti-meron as well as the square network of the domain boundaries (see Appendix A). This mode is analogous to the CCW rotation of the hexagonal SkL found in [27]. Although the Appendix A particularly emphasizes the DW-merons, the topological charge density is concentrated around the center of the unit cell. Interestingly, the “intensity” of the topological charge within DW anti-merons varies depending on the position of the central anti-meron within the square unit cell. The amplitude of the AC field is small enough to exclude any coupling among the merons.

On the other hand, the low-frequency mode, is hard to anticipate. The central anti-merons remain virtually immobile (Figure 6d). A network of domain walls rotate, with each side of the unit cell coming into contact with the central anti-meron in turn. At the point of contact, the structure of the DW anti-meron becomes pronounced and accumulates a large topological charge density; see, e.g., the snapshot in Figure 6d for t=600. Other DW merons are barely discernible, and their topological charges are small compared with the topological charge Q=−1 of a formed bimeron (encircled by the dashed red line). When the central anti-meron moves to the next corner (or rather, the square matrix rotates) and again creates a bimeron structure (now with a different side of the unit cell), the topological charge is first split between two DW anti-merons (e.g., for t=200; such anti-merons are encircled by the dashed red lines) when the central anti-meron is located in the corner of the unit cell. Thus, the low-frequency rotational mode is based on the creation and annihilation of DW merons and enables coupling between anti-merons.

The considered rotational process also provides a hint of the possible scenario of the first-order phase transition between the square meron–antimeron crystals and tilted FM states or elliptical cones.

## 5. The First-Order Phase Transition between 2D Square SkLs and 1D Elliptical Cones

With increasing EPA, the period of the square SkL gradually increases and diverges at $k_{u}\approx -0.765$. Because the inter-meron distances also increase, this results in excessive energy of this phase. The first-order phase transition with the elliptical cone is computed to take place at a somewhat lower EPA of $k_{u}=-0.742$.

The first step of such a transition is to create asymmetry in the balanced position of the central anti-meron when it is attracted simultaneously by the four anti-merons in the domain walls (Figure 7a). In Figure 7b, the central anti-meron is shifted to the left, and as such is attracted by the corresponding boundary anti-meron to reach the minimum of their interaction potential [36].

At the same time, domain-wall merons and anti-merons approach each other, creating prerequisites for their mutual annihilation (Figure 7c). Notice that the domain boundaries bend when the anti-meron acquires the crescent shape typical of bimerons; this additionally facilitates the collapse of DW merons.

After the collapse of excessive merons, the remaining bimerons additionally adjust the value of their dipole moments to reach the minimum of the inter-meron potential (Figure 7d).

Remarkably, in different square unit cells the central anti-merons may be drawn randomly by either side; as a result, the dipole moments of the formed bimerons may form complex bimeron tesselations, so-called bimeron polymers [36]. Because bimerons also attract each other, they may locally assemble into chains or looped clusters, dubbed “roundabouts” and “crossings” in [36]. Figure 7e shows bimerons with mutually perpendicular dipole moments (white arrows). In Figure 7f, bimerons attempt to align into chains with the parallel orientation of dipoles; on the contrary, neighboring chains repel each other. The regions between different bimeron clusters are filled by the elliptical cone or the tilted FM state. In fact, the internal structure of a cone is quite close to the tilted FM state for the chosen control parameters. In this sense, the final state can be viewed as a homogeneous state accommodating isolated bimerons and bimeron clusters.

## 6. Conclusions

In the present paper, I have focused on the essential role of merons arising within domain boundaries between skyrmion cells in chiral magnets with easy-plane anisotropy. Though barely noticeable at the spin distributions and possessing small topological charges as compared with skyrmions, merons nevertheless (i) act as drivers of the structural phase transition between hexagonal and square skyrmion lattices as well as of the first-order phase transition between SkLs and tilted FM states, and (ii) define the dynamic properties of square meron–antimeron crystals.

In particular, I show that merons located in the corners of the hexagonal unit cells merge and consequently “erase” anti-merons with the positive energy density located in between. This process triggers the structural phase transition from the hexagonal to square skyrmion order. Mutual annihilation of merons and anti-merons underlies the subsequent phase transition into the homogeneous state. Anti-merons with opposite polarities are shown to couple and form bimerons, which on the higher level then gather into bimeron networks.

Interestingly, the coupling of anti-merons defines the character of the collective modes induced by the oscillating in-plane fields. In the low-frequency mode, the square-shaped array of domain walls circles around the central anti-meron and lets it successively form a bimeron state with each DW anti-meron. During this process, the coupled DW anti-meron develops a crescent shape and accumulates topological charge, whereas the DW anti-merons within other sides of the unit cell almost decay. In the high-frequency mode, no creation or annihilation of merons is observed; the merons undergo rotations as would be expected for conventional skyrmions.

I argue that the non-trivial findings of the present paper complement previous studies from both fundamental and applied points of view and consider the role of merons from different perspective.

## Figures and Tables

**Figure 1 nanomaterials-14-01524-f001:**
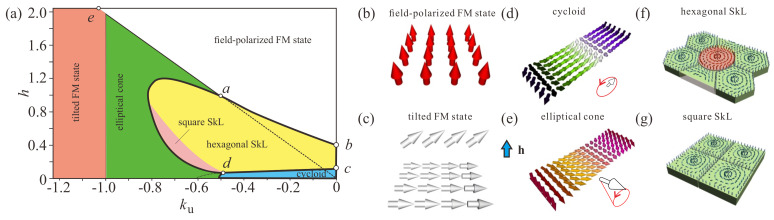
(**a**) Magnetic phase diagram of the solutions for model (Equation 4) with easy-plane uniaxial anisotropy. The filled areas designate the regions of thermodynamical stability of the corresponding phases: white shading—polarized ferromagnetic state (**b**); red shading—tilted ferromagnetic state (**c**); blue shading—cycloidal spiral (**d**); green shading—elliptical cone (**e**); yellow and pink shading—hexagonal (**f**) and square (**g**) skyrmion lattices. The field is measured in units of $H_{0}=D^{2}/A\left|M\right|$, i.e., $h=H/H_{0}$, while $k_{u}=K_{u}A/D^{2}$ is the non-dimensional anisotropy constant; h=0.5 in the following simulations, whereas the anisotropy constant is varied.

**Figure 2 nanomaterials-14-01524-f002:**
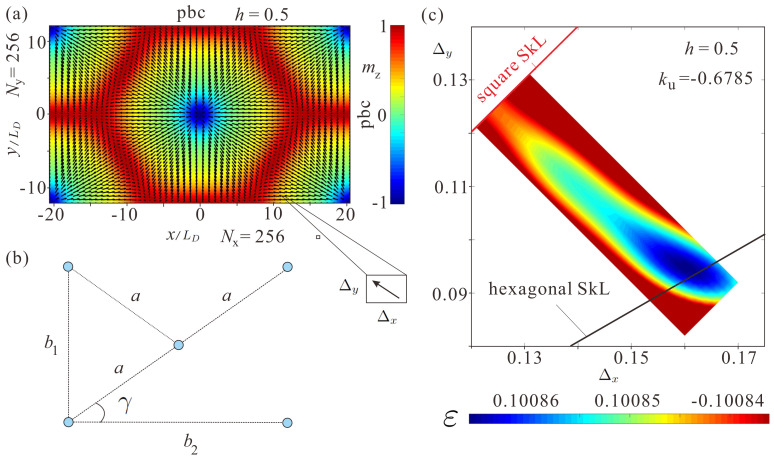
(**a**) Schematics of a computational unit cell corresponding to a distorted (rhombic) SkL. The distribution of the magnetization within skyrmions retains its axisymmetric circular shape. The number of discretization points is equal along *x* and *y*, $N_{x}=256,\;N_{y}=256$. On the contrary, the cell sizes are varied to search for a deformed SkL with the lowest energy density. (**b**) The characteristic geometric parameters of the unit cell, which exhibit the inter-skyrmion distances $b_{1},b_{2}$, and *a* as well as the characteristic angle γ. (**c**) The energy density of distorted SkLs computed by integration of (Equation 4) for different values of lattice spacings and for $h=0.5,\;k_{u}=0.6785$. The well-discernible energy minimum is formed for the skyrmion ordering, which is almost hexagonal. The length scale is measured in units of $L_{D}$ (Equation 5).

**Figure 3 nanomaterials-14-01524-f003:**
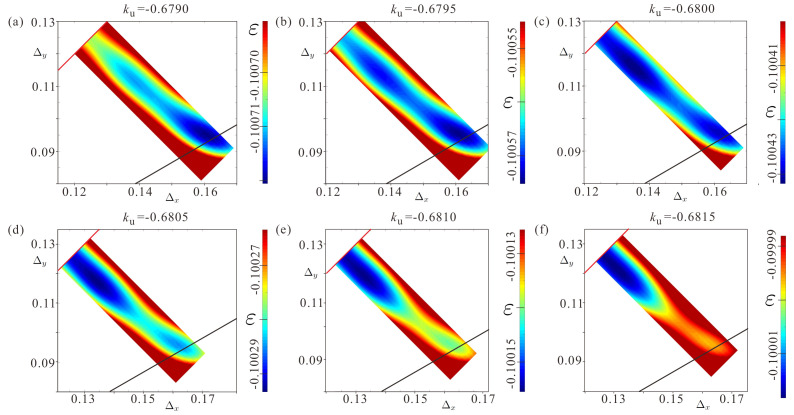
The color plots of the energy densities ε on the $\Delta_{x}$$\Delta_{y}$ plane for increasing values of the uniaxial anisotropy $k_{u}$. Each energy plot features two energy minima: for low values of $k_{u}$, the global minimum corresponds to a hexagonal SkL; for larger $k_{u}$, the global minimum belongs to the distorted (rhombic) SkL on its way towards the square skyrmion order (see (**a**–**f**)).

**Figure 4 nanomaterials-14-01524-f004:**
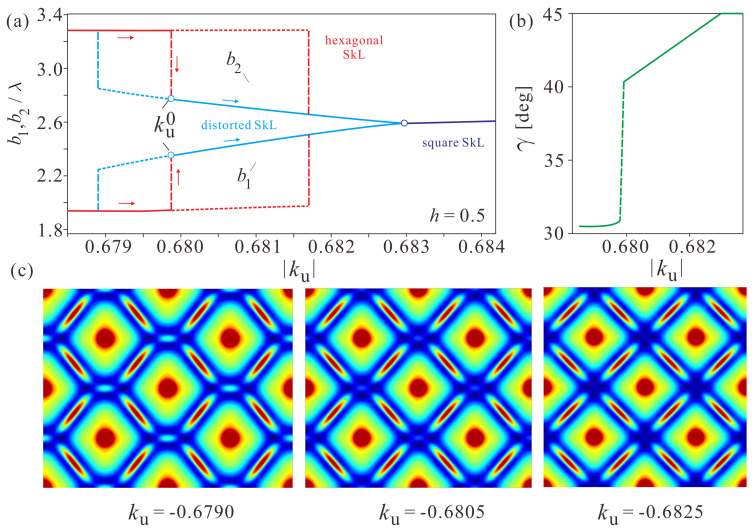
(**a**) Equilibrium parameters of the unit cell, as introduced in Figure 2b, in dependence on the value of the EPA during the structural transition from the hexagonal into the square skyrmion arrangement. The red and blue lines correspond to the hexagonal and distorted SkLs, respectively, with the solid and dotted lines indicating the global and local energy minima. After some value of $k_{u}$, the square SkL (dark blue line) is completely formed. (**b**) The gradual anisotropy-driven change of the angle γ from the value $\gamma \approx 30^{\circ }$ in the hexagonal SkL to the value $\gamma \approx 45^{\circ }$ within the square SkL. (**c**) Color plots of the energy density w(x,y) characterize the gradual evolution of the distorted SkL into a square one.

**Figure 5 nanomaterials-14-01524-f005:**
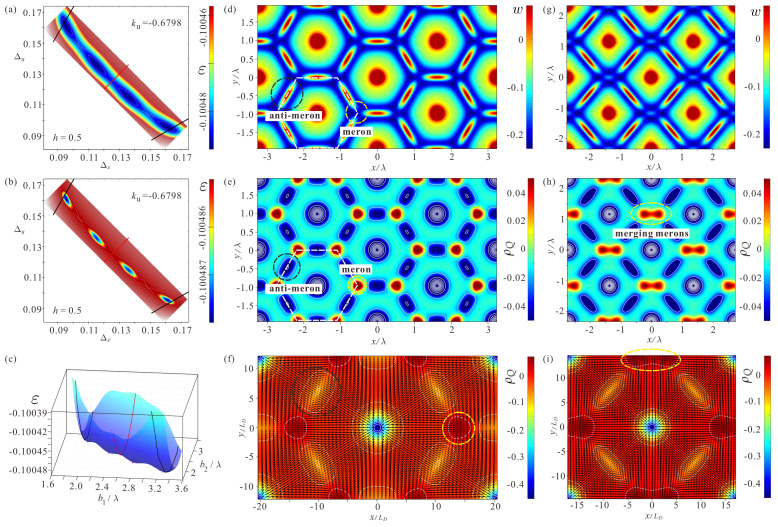
(**a**) The color plot of the energy density ε on the $\Delta_{x}\Delta_{y}$ plane. The black and red lines show the parameters for the hexagonal and square SkLs, respectively. The legend highlights the energy range $\varepsilon \in [\varepsilon_{min},\varepsilon_{min}+3\times 10^{-5}]$. (**b**) The same energy plot, now shown in the range $\varepsilon \in [\varepsilon_{min},\varepsilon_{min}+3\times 10^{-6}]$ to prove that all four energy minima share the same depth. (**c**) The energy density of distorted skyrmion orders plotted as a surface. The hexagonal SkLs (highlighted by the black curves) almost reach the global energy minima. The parameters for the square SkL constitute a red curve with the minimum being a saddle point, i.e., there is no solution for the square SkL for this value of the EPA. The pink dots show the actual energy minima of the system. (**d**,**g**) Contour plots for the energy density distributions w(x,y) within the hexagonal and distorted SkLs, respectively, for $h=0.5,\;k_{u}=-0.6798$. In both graphs, the legends highlight the same energy range $\left[w_{min},w_{min}+0.3\right]$. (**e**,**h**) Color plots of the topological charge density $\rho_{Q}$ in both skyrmion arrangements. The legends “zoom” the interval [−0.05,0.05]. (**f**,**i**) Distributions $\rho_{Q}\left(x,y\right)$, with the legends exhibiting the intervals $\left[\rho_{Q}^{min},\rho_{Q}^{max}\right]$. The black arrows show the projections of the magnetization vectors onto the xy plane.

**Figure 6 nanomaterials-14-01524-f006:**
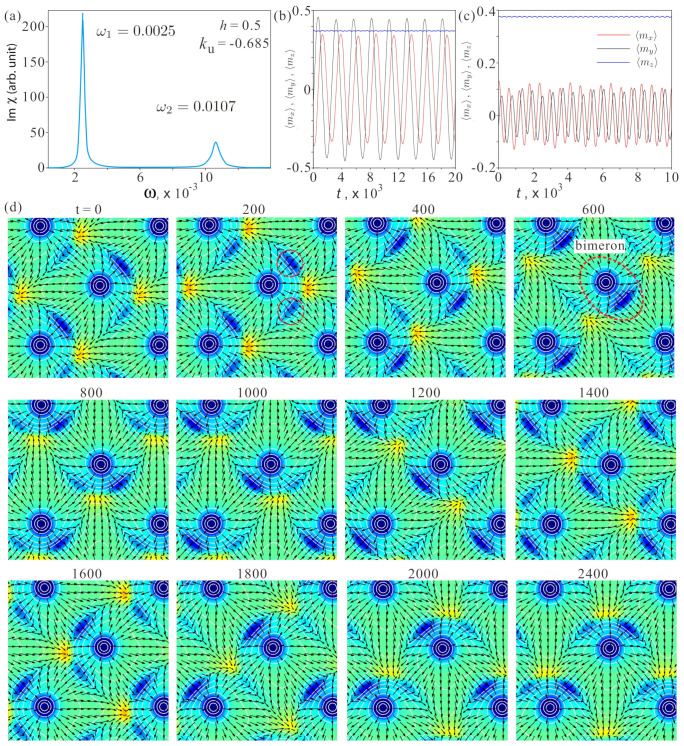
(**a**) Imaginary part of the in-plane dynamical susceptibility for $k_{u}=-0.685,\;h=0.5$, exhibiting two resonance frequencies. (**b**,**c**) Calculated time evolutions of the averaged magnetization components $\langle m_{x}\rangle ,\;\langle m_{y}\rangle ,\;\langle m_{z}\rangle$ in both spin–wave modes. (**d**) Spin dynamics within the low-frequency mode characterized by snapshots of the topological charge density (see text for details).

**Figure 7 nanomaterials-14-01524-f007:**
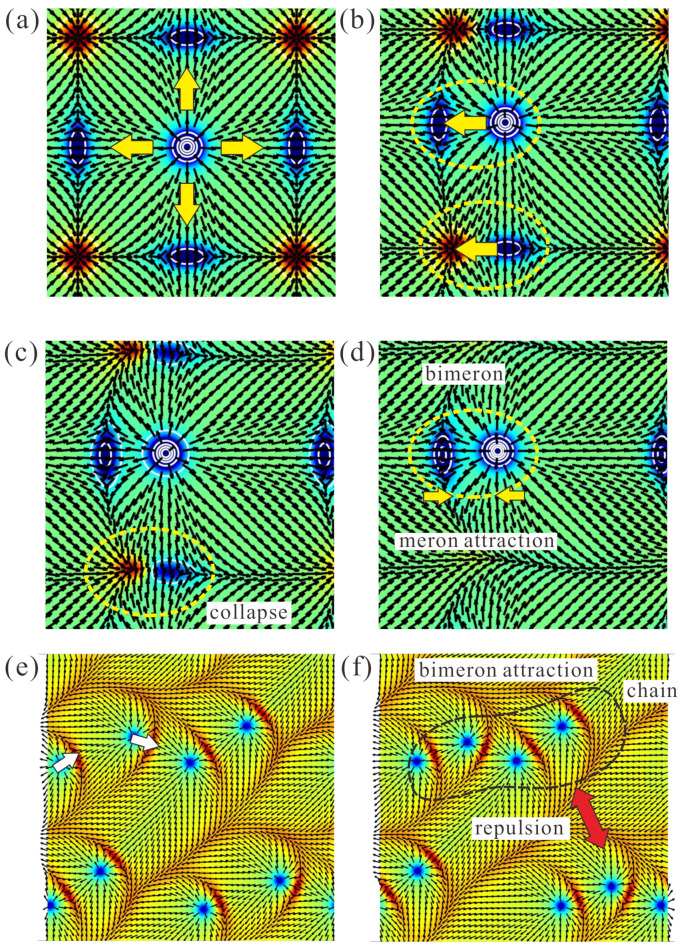
Mutual transformation between the square SkL and elliptical cone (tilted FM state) occurring via collapse and coupling of merons. The internal structures of the states in (**a**–**d**) are shown as color plots of the topological charge density. The color plots in (**e**,**f**) exhibit the $m_{z}$-component of the magnetization. The elliptical cone in these figures contains a network of attracting bimerons (see text for details). The dashed yellow ellipse in (**c**) highlights the pair of oppositely charged merons, which are about to collapse. The yellow line in (**d**) encircles a stable bimeron state formed out of two negatively charged merons with the opposite polarity. White arrows in (**e**) signify dipole moments of bimerons. The dashed black line in (**f**) highlights the area, in which bimerons try to align into a straight chain with parallel dipole moments. Neighboring chains repel each other.

## Data Availability

Data are contained within the article and Appendix A.

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
