# Peer review of "Meron-Mediated Phase Transitions in Quasi-Two-Dimensional Chiral Magnets with Easy-Plane Anisotropy: Successive Transformation of the Hexagonal Skyrmion Lattice into the Square Lattice and into the Tilted FM State"

_nanomaterials, 2024, doi:10.3390/nano14181524_

Round 1

Reviewer 1 Report

Comments and Suggestions for Authors

This article addresses the role of merons (half-skyrmions) in chiral lattices.  It shows that merons can be used to understand: (i) the phase transition between square and hexagonal skyrmion lattice; (ii) spin-wave resonances in the skyrmion lattice; and (iii) the phase transition between the square skyrmion lattice and the elliptical cone phase.

The results are interesting and should be published.  However, the quality of writing could be improved in several places (see comments on language below).

Comments on the Quality of English Language

The quality of writing is mostly good but could be improved in several places:

P1 The abstract is very long and does not succinctly summarise the most important results.  This abstract is 374 words long whereas 100-150 words is typical for a Nanomaterials article.

P2 L48 "where dx_k=d/dx_k" creates confusion -- (1) is unambiguous and requires no explanation.

P2 "the modulated phases and solitons are expected to be truly three-dimensional" this paragraph seems to be irrelevant.  The results of the article concern 2D systems, not 3D systems.

P2/3 The acronym "EPA" is defined in a figure caption but should be defined in main text (e.g. on P2 line 90).

P3 L120 "and anti-merons -- by two skyrmion cells" this hyphen is unnecessary

P6 L170 says that when b1=b2 and gamma=45 degrees, a=b1. But Fig 2(b) suggests that a=b1/sqrt(2) in this situation.

P9 L267 is kappa the same as chi?

P10 L279 "As could be anticipated, the high-frequency mode represents a CCW rotation..."  What is the meaning of the acronym "CCW"?  And why is this behaviour anticipated?  The description of this mode is unclear e.g. it is hard to guess what is meant by "twinkling intensity".

P11 L323 "repulse each other" should be "repel"

P12 L344 "array of domain walls circles ... and let it form a bimeron state" should be "lets"

P12 L346 "whereas DW anti-merons" should be "whereas the DW anti-merons"

Author Response

Please find an attached pdf-file

Reviewer 2 Report

Comments and Suggestions for Authors

The paper presents a numerical study of chiral magnetization configurations due to Dzyaloshinskii Moriya interaction and of their collective oscillation modes as well. Here are some remarks: 

1/ mâ‚š are the components of the unit vector parallel to the magnetization. The exchange energy is A Σâ‚™,â‚š (∂mâ‚š/xâ‚™)², the DMI energy is D Σâ‚š (mâ‚š∂m₃/xâ‚š - m₃∂mâ‚š/xâ‚š), the Zeeman energy is - M Σâ‚š Hâ‚š mâ‚š and the anisotropy is -K m₃². With the characteristic length L = A / D,  the non dimensional coordinates are ξâ‚™ = xâ‚™ / L, the exchange energy is A/L² Σâ‚™,â‚š (∂mâ‚š/∂ξâ‚™)² and the DMI energy is D/L Σâ‚š (mâ‚š∂m₃/∂ξâ‚š - m₃∂mâ‚š/∂ξâ‚š). With the characteristic surface energy E = D/L, the exchange energy is E (A/(DL)) Σâ‚™,â‚š (∂mâ‚š/∂ξâ‚™)², the DMI energy is E Σâ‚š (mâ‚š∂m₃/∂ξâ‚š - m₃∂mâ‚š/∂ξâ‚š), the Zeeman energy is - E ( M L/ D) Σâ‚š Hâ‚š mâ‚š and the anisotropy is - E ( KL/D) m₃². As L = A/D, the exchange energy is E  Σâ‚™,â‚š (∂mâ‚š/∂ξâ‚™)², the DMI energy is E Σâ‚š (mâ‚š∂m₃/∂ξâ‚š - m₃∂mâ‚š/∂ξâ‚š), the Zeeman energy is - E ( M A/ D²) Σâ‚š Hâ‚š mâ‚š  and the anisotropy is - E ( KA/D²) m₃². Thus it is convenient to set hâ‚š = Hâ‚š ( M A/ D²) and k = KA/D² and to write the sum of these energies E ( Σâ‚™,â‚š (∂mâ‚š/∂ξâ‚™)² + Σâ‚š (mâ‚š∂m₃/∂ξâ‚š - m₃∂mâ‚š/∂ξâ‚š)  - Σâ‚š hâ‚š mâ‚š - k m₃² ). The expression given in the paper for k is not correct.

2/ The energy due to dipolar interaction is not mentioned in the paper. This energy highly influences the magnetization configurations in all the real devices. Is it included in the anisotropy? This approximation should be at least mentioned and also discussed. Moreover, it also influences the frequency of the collective oscillation modes. What does the author expect if the real dipolar energy is taken into account. How does it influence the transition from hexagonal to square lattices? How does it change the frequencies? To support my concerns, we can consider a propagating spin wave in a saturated ultra thin layer in the so-called Damon Eshbach configuration i.e. the in-plane applied field is perpendicular to the wave vector. If the dipolar interaction was included in the anisotropy, then the frequency f would be given by the relation (2π f / γ + 2 D Ï° /M ) = sqrt ( (H + 2 A Ï°² /M ) ( H + 2 A Ï°² /M + 4π M - 2K/M) ) where Ï° is the wave vector. In a matter of fact, the correct relation is (2π f / γ + 2 D Ï° /M ) = sqrt ( (H + 2 A Ï°² /M + 4π M ( 1 - F(Ï° t) ) ( H + 2 A Ï°² /M + 4π M F(Ï° t)- 2K/M) ) where t is the layer thickness and  F(u) =  (1 - exp( |u| ) ) / |u|. In all the real devices, the thickness is not vanishing. In the case of a skyrmion, the characteristic length is R, the radius of its core. Thus the validity of the approximation depends on t/R …

Author Response

Please find an attached pdf-file

Reviewer 3 Report

Comments and Suggestions for Authors

This manuscript has studied the phase transitions between hexagonal and square skyrmion lattices (SkLs) as well as between the square skyrmion lattices and 1D elliptical cones theoretically. Moreover, two resonance peaks result from the complex dynamics of domain-wall merons. The processes of the phase transitions have been comprehensively investigated. The results are important for understanding the mechanisms of phase transitions between different skyrmion lattices. The manuscript is well written, but some minor issues should be carefully addressed as below:

(1)   In the formula (2), page 2, the second term L(y)x,z should be changed to L(y)z,x.

(2)   In Line 170, page 6, for the square SkL, it should be “b1=b2=√2a”, not “b1=b2=a”.

(3)   For the square SkL in Fig. 5(g-i), each corner meron is shared by four neighboring square unit cells, thus there is 1/4×4=1 meron in one unit cell of the square SkL. However, for the hexagonal SkL in Fig. 5(d-f), it seems that each corner meron is shared by three neighboring hexagonal unit cells, thus there should be 1/3×6=2 merons in one unit cell of the hexagonal SkL. Why does the author claim that each corner meron is shared by six neighboring unit cells?

(4)   Is the Fig. 5(c) the surface plot of the Fig. 5(a) or Fig. 5(b)? If so, why are the ranges of the x and y axes (Δx and Δy) in Fig. 5(c) different from those in Fig. 5(a)/(b)?

(5)   In Line 307, page 10, the central anti-meron is shifted to the left in Fig. 7(b), not “central meron”.

(6)   The full names of the abbreviations “LLG” and “CCW” should be provided for the first use.

(7)   There is a typo in Fig. 7 caption. The “illiptical” should be “elliptical”.

(8) The manuscript mainly studied the phase transitions between hexagonal and square skyrmion lattices and between the square skyrmion lattices and the tilted ferromagnetic state. "Phase transitions between square/hexagonal skyrmion lattices and tilted FM states" in the manuscript title will cause misunderstanding. Please revise it.

Author Response

Please find an attached pdf-file

Round 2

Reviewer 2 Report

Comments and Suggestions for Authors

the author addressed the issues raised by the first version